# Monitoring *Bemisia tabaci* (Gennadius) (Hemiptera: Aleyrodidae) Infestation in Soybean by Proximal Sensing

**DOI:** 10.3390/insects12010047

**Published:** 2021-01-09

**Authors:** Pedro P. S. Barros, Inana X. Schutze, Fernando H. Iost Filho, Pedro T. Yamamoto, Peterson R. Fiorio, José A. M. Demattê

**Affiliations:** 1Civil Engineering College, University Federal of Uberlândia, Monte Carmelo Campus, Monte Carmelo, Minas Gerais 38500-000, Brazil; 2Department of Entomology and Acarology, University of São Paulo, Piracicaba, São Paulo 13418-900, Brazil; inana.schutze@usp.br (I.X.S.); fernandohiost@usp.br (F.H.I.F.); pedro.yamamoto@usp.br (P.T.Y.); 3Department of Biosystems Engineering, University of São Paulo, Piracicaba, São Paulo 13418-900, Brazil; fiorio@usp.br; 4Department of Soil Science, University of São Paulo, Piracicaba, São Paulo 13418-900, Brazil; jamdemat@usp.br

**Keywords:** glycine max, sampling, pest management, spectroradiometer

## Abstract

**Simple Summary:**

The whitefly *Bemisia tabaci* has become a primary pest in soybean fields in Brazil over the last decades, causing losses in the yield. Its reduced size and fast population growth make monitoring a challenge for growers. The use of hyperspectral proximal sensing (PS) is a tool that allows the identification of arthropod infested areas without contact with the plants. This optimizes the time spent on crop monitoring, which is important for large cultivation areas, such as soybean fields in Brazilian Cerrado. In this study, we investigated differences in the responses obtained from leaves of soybean plants, non-infested and infested with *Bemisia tabaci* in different levels, with the aim of its differentiation by using hyperspectral PS, which is based on the information from many contiguous wavelengths. Leaves were collected from soybean plants to obtain hyperspectral signatures in the laboratory. Hyperspectral curves of infested and non-infested leaves were differentiated with good accuracy by the responses of the bands related to photosynthesis and water content. These results can be helpful in improving the monitoring of *Bemisia tabaci* in the field, which is important in the decision-making of integrated pest management programs for this key pest.

**Abstract:**

Although monitoring insect pest populations in the fields is essential in crop management, it is still a laborious and sometimes ineffective process. Imprecise decision-making in an integrated pest management program may lead to ineffective control in infested areas or the excessive use of insecticides. In addition, high infestation levels may diminish the photosynthetic activity of soybean, reducing their development and yield. Therefore, we proposed that levels of infested soybean areas could be identified and classified in a field using hyperspectral proximal sensing. Thus, the goals of this study were to investigate and discriminate the reflectance characteristics of soybean non-infested and infested with *Bemisia tabaci* using hyperspectral sensing data. Therefore, cages were placed over soybean plants in a commercial field and artificial whitefly infestations were created. Later, samples of infested and non-infested soybean leaves were collected and transported to the laboratory to obtain the hyperspectral curves. The results allowed us to discriminate the different levels of infestation and to separate healthy from whitefly infested soybean leaves based on their reflectance. In conclusion, these results show that hyperspectral sensing can potentially be used to monitor whitefly populations in soybean fields.

## 1. Introduction

According to the United States Department of Agriculture (USDA) [1], world soybean (*Glycine max* (L.) Merril) production in the 2018/2019 season was 361.06 million tons. Brazil is projected to be the largest producer of soybeans in the world by 2021 [2]. In Brazil, the 119.70 million tons harvested in 2018/2019 were grown in around 35.90 million hectares, meaning that the average yield was 3.26 tons per hectare. In the 2019/2020 season, the average Brazilian yield is projected to increase by 3.9% and a 2.7% increase in the area [3]. Although this is the main product of Brazilian agribusiness today, representing a quarter of the gross production value of agriculture in Brazil [4], the monoculture in wide fields has consequences, such as greater vulnerability to insect pests, causing a reduction in yield [5].

Therefore, knowing and monitoring the main pests present in the soybean ecosystem, using a variety of sampling methods, is extremely important for the decision-making to be taken at the right time, avoiding yield losses [6]. The occurrence and damage of the whitefly *Bemisia tabaci* (Gennadius, 1889) (Hemiptera: Aleyrodidae) in soybean fields are alarming; in high densities, it can cause losses between 12% and 30% in yield [7]. In addition, this pest is able to tolerate the action of some insecticides, with a rapid selection of resistant populations [8].

There are parameters that allow decisions to be taken at the right time, resulting in better control like the economic injury level (EIL) [9] and the economic threshold (ET) [10], the pest population at which actions should be taken, prevent such population from reaching the EIL [11].

These levels are already established for the main pests that infest soybean plants [12]. However, for some species that became more important recently, such as whiteflies (Hemiptera: Aleyrodidae), spider mites (Acari: Tetranychidae) and even pod-eating caterpillars (Lepidoptera: Noctuidae), the EIL and ET are still being investigated [13]. To acquire data and calculate these levels, crop fields must be regularly checked for pests. In soybean fields, the sampling method most used for monitoring insects that inhabit the aerial part of the plants is the beat cloth, initially introduced in the United States [14], this method was later modified and introduced in conditions of Brazilian agriculture [15,16].

The sampling methods currently in use are challenging, considering the vast extension of soybean fields in the Midwest region of Brazil [17]. In addition, they are time-consuming and expensive due to the necessity of workers scouting the field [18,19]. Moreover, there is still a lack of reliable sampling methods for some species, especially those too small to see with the naked eye or the ones that inhabit the soil. This scenario promotes the implementation of remote or proximal sensing technologies and their benefits, especially the potential time saved by automatizing crop monitoring [19,20,21].

Recently, proximal sensing has been gaining adherents in Brazil, especially in sugarcane fields [22,23], and later in grain fields, such as soybean [24]. Although precision tools currently are used more for planting/sowing operations, fertilization and weed control, there is a growing interest of researchers in providing tools to be used in insect and plant pathogen (disease) management [25]. The main difference between those operations and insect/disease management is that the former is based on collecting data from the soil, while the latter is based on collecting spectral data from plants.

To obtain spectral data from plants, it is necessary to understand that, when light reaches the leaves (canopy), part of that energy is reflected back to the observer. The amount of energy reflected at each wavelength is called the reflectance spectrum, sometimes shortened to spectrum or reflectance. Reflectance depends on the properties of the leaf surface and its internal structure, as well as the concentration and distribution of biochemical components [26,27]. Moreover, abiotic stresses, such as herbivory by arthropod pests, induce physiological responses in plants that impair their ability to perform photosynthesis, leading to changes in leaf reflectance in some parts of the spectrum. On this matter, most studies refer to the 400–2500 nm range, especially with hyperspectral sensors [28]. Hence, advanced sensing technologies can be used to detect changes in reflectance from soybean plants as a non-invasive monitoring method [17].

Remote/proximal sensing has been used to detect stress caused by arthropod herbivory in a variety of plant species, such as maize [29], soybean [6], rice [30,31], wheat [32], peach trees [18,33], cotton [34] and potato [35]. The most promising results were achieved in studies based on hemipteran pests because their feeding activity (sucking) indirectly affects the infested plants’ physiology, and therefore, their reflectance profiles [36]. In most of these studies, infested and non-infested plants were discriminated against with good accuracy.

In the search for responses more detailed than infestation vs. non-infestation, a variety of approaches have been used to analyze reflectance data from plants infested with different pest densities. In general, the best correlation indexes were achieved when infestation levels (classes, not the absolute number of insects per plant) and narrow-band wavelengths (not individual wavelengths) were compared [37]. However, it is still necessary to study how different analytical approaches interfere in the quality and usability of such information remotely extracted from infested plants. 

Hence, it can be said that one of the biggest challenges regarding hyperspectral remote sensing is the analysis of a large number of bands. This analysis is complex and time-consuming, using special algorithms to select a group of bands sensitive to arthropod infestation in each plant species [28]. According to Hair et al. [38], currently, one of the most used statistical procedures to reduce the amount of data without losing important information is multivariate analysis. One example of multivariate analysis is the discriminant analysis that is done with the objective of separating the observations into groups [39]. In addition, classification analysis is done to assign observations whose group memberships are unknown to the established groups based on *p* measured values [29]. This association is only possible if part of the observations from each group is previously available. Thus, this study aimed to develop models to discriminate the levels of whitefly infestation in soybean fields, using hyperspectral proximal sensing.

## 2. Material and Methods

### 2.1. Local

The bioassay was carried out in the experimental field at the College of Agriculture “Luiz de Queiroz”, from the University of São Paulo, located in Piracicaba, Sao Paulo state, Brazil. The area is located at the following coordinates: Datum (SIRGAS 2000): 22°42′16″ S Lat.; 47°37′23″ W Long.; approximated altitude 532 m.

The climate is humid subtropical climates, with dry winter and hot summer (CWa), according to Köppen classification [40]. The average year pluviosity is 12,800 mm, and the average temperature is 22 °C, with the average temperature in the hottest month of 25 °C and 18 °C in the coldest month.

Conventional soybean, variety BRS 232, was sown on November 28th, 2018, in an area of 1.5 hectares. The field was tilled and fertilized with nitrogen, phosphorus and potassium, following the standard procedures used by the grower in the cultivated area. The soil is classified as dystrophic red-yellow latosol.

### 2.2. Insect Rearing

The rearing of whitefly, *B. tabaci* biotype B, started from a population acquired at the Agronomic Institute of Campinas. The population is maintained in kale plants and kept in a greenhouse covered with an anti-aphid screen [41]. The plants are replaced in the greenhouse when necessary in order to keep the insect population adequate for the development of bioassays.

### 2.3. Bioassay

The bioassay began on December 13th, 2018, when the soybean plants reached the phenological stage V3 (third node, two fully expanded trifoliate) [42]. The treatments were distributed in a randomized block design, made of four blocks and four treatments (low, medium, high and control) consisting of different *B. tabaci* infestations, totalizing 16 experimental units. Each experimental unit consisted of a cage (2.0 m long, 1.7 m large, and 1.6 m high) set up over the crop in the field. The cages were supported by bamboo poles and covered with an anti-aphid screen that allows airflow and prevents infestation by unwanted arthropods. The cages were installed on December 12th, 2018, 2 m apart from each other, and comprised about 75 plants each one.

On December 19th, 2018, the cages were manually infested, releasing in each cage one pot with one kale plant and the amount of insect corresponding to each treatment. The treatments were: 1—control (no insects); 2—low (approximately (ca.) 300 adults); 3—medium (ca. 600 adults); and 4—high (ca. 1200 adults), the number of adults released was intended to reach densities of nymphs enough to differentiate the treatments from each other in a period of weeks. The insects continued feeding on the plant along the soybean crop cycle.

### 2.4. Data Collection

To collect reflectance data, ten leaflets from the middle third of the soybean plants were collected from each cage and stored in plastic bags with identification tags, a total of 160 leaflets per collection. The leaflets were collected on January 10th, 2019; January 17th, 2019; January 24th, 2019; January 31st, 2019; February 7th, 2019; February 14th, 2019; February 2st1, 2019; and February 28th, 2019. Then, the samples were taken to the laboratory in a thermal box with ice cubes to maintain the turgidity of the leaves during the collection of spectral data.

Spectral data were collected from each leaflet using a spectroradiometer (FieldSpec 3, Analytical Spectral Devices, Boulder, CO, USA). This sensor operates in the spectral range of 350–2500 nm, with a spectral resolution of 1.4 nm in the range of 350–1050 nm and 2 nm in the range of 1051–2500 nm. The sensor was connected to the ASD Leaf Clip accessory (Analytical Spectral Devices, Boulder, CO, USA), designed for nondestructive spectral measurements, without interference from external light, minimizing errors associated with diffuse light. This accessory has a halogen light source (4.5 W) with an incidence light of 45° for the sample, which allows the measurement of the directional reflectance of the light directly from the sample.

A Barium plate that reflects 100% of the light was used as a reflectance standard. The spectral data were stored by the system for posterior determination of the samples’ reflectance factor, which was multiplied by the readings of each sample.

The central region of each leaflet was evaluated in a circle of 2.1 cm in diameter (area of 3.5 cm^2^), resulting in one spectral sample per leaflet.

There was a total of eight sampling dates. At each sampling date, 10 leaflets from each one of the 16 cages were sampled, in a total of 160 spectral samples. All leaflets were collected in an interval of less than an hour to allow comparison. After obtaining the spectral data, the nymphs of each leaflet were counted in a stereoscopic microscope (40× magnification) to obtain the infestation data.

The meteorological data were obtained from the weather station of the University of Sao Paulo [43]. The information collected from the website was the maximum, average and minimum temperature (°C) and precipitation (mm).

### 2.5. Data Analysis

A large amount of data in a spectral curve makes it difficult to group samples into different classes based on visual criteria alone. In addition, according to Bauriegel et al. [44], the reflectance in the same spectrum presents high collinearity, producing a large number of redundant information. Therefore, a multivariate analysis was used to reduce the dimensionality of the data and to determine the effects of treatments more clearly.

According to Nansen and Elliot [28], the use of multivariate statistics is the best tool to interpret the spectral behavior of vegetation under stress, allowing interpretations that would not be possible using univariate statistics.

The software XLSTAT [45] was used to analyze the data matrix of 1950 wavelengths (range of 450–2400 nm). A discriminant analysis was carried out to develop and validate a method to determine infestation levels using spectral data. Thus, the spectral curve was condensed into a single point, along with its discriminatory value. By calculating the average value of discriminant points from a group, we obtain the group’s average, known as centroid. The verification of the significance of the discriminant functions is a generalized measure of the distance between the groups’ centroids. Therefore, if the distribution of the discriminating scores in each group shows little overlap, the discriminating function separates the groups well [38].

To do the discriminant analysis, a simulation was carried out with 70% of the samples to generate a discriminant model, which was tested in the 30% remaining samples. The ratio selection was random, as well as the selection of which samples would be part of the model (70%) or the test (30%).

## 3. Results and Discussion

In the discriminant models generated for each of the eight sampling dates, some bands were observed more frequently than others (Figure 1). The frequency of distribution of the bands with the greatest weight in all the eight models generated can be observed. Some bands in the visible region (461, 469, 510, 520 and 673 nm), near-infrared, NIR region (703, 722 and 732 nm), and shortwave infrared (SWIR) (1360, 1426, 1713, 1819 and 1842 nm) were observed in two of the eight discriminant models. The individual band 1831 nm was observed in three of the eight models.

Regarding the differentiation between treatments (low, medium, high infestations, and control) based on the discriminant analysis, the best results were achieved on 31 January, with 75.50% accuracy (Figure 2), when the soybean was in the reproductive stage R4. Such accuracy was obtained in the cross-validation analysis, where part of the samples was provided to the machine as a learning set, and the rest of the samples (validation set) were classified by the machine based on the learning set. In this case, accuracy (%) means how much treatment classification by the machine was similar to the real treatments in the field.

By analyzing the infestation data together with the meteorological data, it is possible to observe that the period was dry and hot (Figure 2), boosting the development of whitefly populations in the field.

Therefore, only the data collected on 31 January was used for a more detailed analysis. Evaluating the spectral curves that represent the average reflection of each infestation level, we could observe a difference in the reflectance intensity (Figure 3). More specifically, the high level of infestation showed greater reflectance across the analyzed electromagnetic spectrum compared to the other levels.

The water bands, highlighted in Figure 3, occur when the energy in these wavelengths interacts with OH of the water molecules, causing a vibrational effect. The effect absorbs the energy of this wavelength, not reflecting it. With no reflection, the absorption feature occurs. In these bands, it is observed that the most infested area has less water. This is indicative of water stress precisely because the plant is not managing to keep the turgidity; this could be related to the infested leaves not being able to transport water due to damages caused by *B. tabaci* feeding, affecting xylem on the vascular bundles, where phloem is also located [46].

In the range of the NIR 800–1000 nm, the band related to the leaf structure, a higher intensity was observed in the most infested plants. This was a notorious fact because the better the structure is, the greater the reflectance intensity is expected. However, an anatomical investigation showed that, although *B. tabaci* impacts the leaf anatomy, they occur on the abaxial surface, where the vascular bundles are located, while we used the adaxial surface to perform the data collection. The differences found here are more related to the physiology of the leaf rather than its anatomy [46].

High densities of *B. tabaci* causes the occurrence of honeydew. In the nymphal stage, they excrete a high volume of this sugar-rich watery fluid [47], which is a substrate for the development of fungi of the genus *Ascomycete* that produces the symptom known as sooty mold. This symptom turns the foliar surface to black, causing more solar radiation to be absorbed, resulting in burns and falls. This pathosystem can be limiting for photosynthesis and, therefore, reduce plant production. This situation was observed in the visible region (Figure 4), where the high level of infestation presented higher reflectance intensity that is directly related to photosynthetic pigments. With the lower photosynthesis, the plant does not absorb wavelengths at the blue and red ranges, and thus, reflection gets higher.

The results shown in the spectral curves in the wavelengths 450–750 nm indicate low reflectance (around 10%), with a slight increase in the region correspondent to green light (550 nm) (Figure 4). The reduction in reflectance is often associated with the absorption of foliar pigments due to the presence of chlorophyll. In the spectral region correspondent to blue light, the absorption occurs near the wavelength 460 nm and is related to the presence of xanthophyll, carotenes, and chlorophyll pigments *a* and *b*. In the red-light region, chlorophyll acts as absorbing energy near 645 nm [47].

Thus, it is possible to observe (Figure 4) that the treatment with the least photosynthetically activity (high infestation) presented higher reflectance in all the visible region spectrum. This may have occurred because of how whitefly infestation affects the plant physiology, altering water balance, photosynthesis, chlorophyll content and metabolites associated with physiological stress [48,49].

Most of the processes mentioned, which alter the leaves’ and plants’ physiology are observable in the spectral signature of such plants (Figure 4). However, advanced statistical tools are required to know how much each process is correlated to each wavelength. Moreover, these analyses are necessary to identify which wavelengths are more significant for each of these processes.

Hence, observing the discriminant analysis (Figure 5) in the reproductive stage R4, where each point of the graphic represents one spectral curve from the leaflets collected from the cages, we can see that the first function explains 60.86% of the data variability, while the second function explains 24.15%.

After performing the discriminant analysis (DA) with 70% of the samples, the next step was the validation of the model, with 30% of the samples left. With the discriminant function obtained, the cross-validation was performed, and the samples were identified in their correspondent infestation level, with a total accuracy rate of 75.48% (Table 1). The plants with medium infestation were classified more accurately (85.71%), while most of the errors in classification occurred in plants with a low infestation (69.57%). The difficulty with this discrimination is the lack of visible symptoms early in the season and connecting the factors causing the biophysical/chemical changes in the plants. At the date when the data were collected, the average number of nymphs per leaflet in the medium infestation was two times the number of nymphs in the low infestation treatment, which could make it more difficult to separate the low infestation from the control group. Soybean plants have a good water compensation [50], which could provide a good response against sucking pests until certain levels; by raising the level of whitefly infestation, the amount of water consumed by this pest which feeds on the phloem vessels also increases, allowing to see a better distinction between the treatments. On the other hand, as the number of whiteflies present in the leaves goes up, the occurrence of the sooty mold also increases, which harms a more accurate reading of the data due to a more complex scenario, whereas the average number of nymphs per leaflet in the high infestation was almost eight times higher than the medium infestation.

Analyzing all the bands selected, we have 17 wavelengths, 5 in the visible region (450–682 nm), 6 in the near-infrared region (716–1167 nm), and 6 in the shortwave infrared region (1321–2265 nm).

Blue wavelengths (450 and 499 nm were selected in DA) are strongly influenced by chlorophyll absorption, along with carotenoid absorption features present in the 450–499 nm region. Carotenoids have proven important for the discrimination of senescent leaves when the decay of chlorophyll and the diminishing of the strong chlorophyll-absorption feature reveal the carotenoid absorption feature [46].

The red edge (682, 716, 739, 748 nm were selected in DA) encompasses the region from the red reflectance minimum around 680 nm to the near-infrared (NIR) shoulder at approximately 780 nm. This region indicates a sharp increase in reflectance from the visible (VIS) to NIR regions associated with strong chlorophyll absorptions and internal leaf structure [46].

The focused shortwave infrared (FSWIR) (2265 nm was selected in DA) has the lowest average band selection rate, with its highest selection at bin 2250–2299 nm most likely associated with the weak absorption features of cellulose and lignin present at 2270 nm [46].

Thus, the intervals that had higher representativity were visible and SWIR. One possible reason for this result is the fact that these regions are related to photosynthesis, light absorption for this process, and water absorption. Moreover, the feeding behavior of whitefly can affect all the three processes mentioned above. Both nymphs and adults feed on phloem using their stylets [41].

Phloem is a vegetal tissue made of sieve elements and sclerenchyma and parenchyma cells. The main functions of these parts are to transport photoassimilates (organic compounds produced by photosynthesis). These functions are related to the wavelengths mentioned, and best represented the interaction between whitefly infestation and the spectral curves. This is due to the fact that this tissue is the most affected by this pest.

To sum up, our results show that, in the conditions tested, it is possible to separate healthy and whitefly infested soybean plants based on foliar reflectance. In addition, we can separate the levels of infestation (low, medium and high) with good accuracy, using classification analysis. The uniqueness of this technique is related to the plant data acquisition with remote sensors, which could be used in commercial fields to improve pest monitoring in the future. In the specific case of whitefly, this approach is extremely relevant because of the difficulty in visually monitoring very small insects in large fields. Hence, the use of monitoring systems based on plant reflectance is a very promising tool.

These results show that future research needs to be done in larger areas and natural infestation levels to validate the sampling technique proposed in this study, using other sensors and conditions. More specifically, it is necessary to understand the spectral behavior of soybean plants out of the experimental cages used in this study, as well as to analyze the efficiency of sensors attached to terrestrial or aerial platforms. After being validated, this technique can be used to increase the implementation of IPM programs, to determine where and when control methods are required for managing the pest.

Hence, the translation of the spatial, spectral, and radiometric information obtained by hyperspectral spectroradiometers into multispectral sensor resolution demands much attention, being one more feasible way of taking this information into the crop fields in the present.

## 4. Conclusions

It is possible to separate healthy and whitefly infested soybean leaves based on their spectral reflectance. In addition, the results obtained by the discriminant analysis of the hyperspectral data showed a clear distinction between the different levels of infestation. Finally, the NIR and SWIR were the most important for the model, as they are directly related to photosynthesis and water content in the leaves.

## Figures and Tables

**Figure 1 insects-12-00047-f001:**
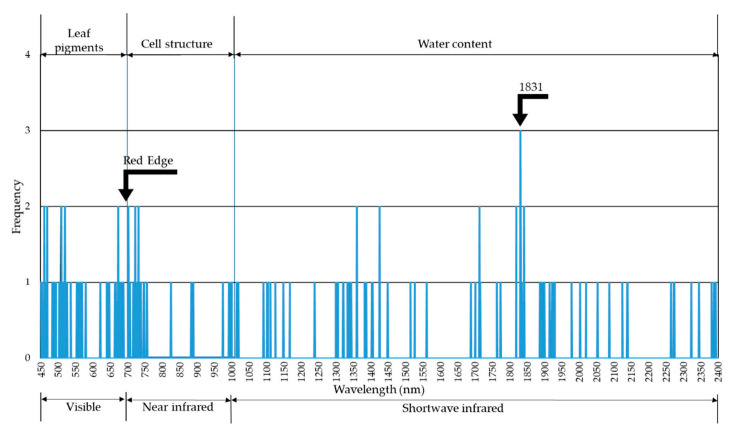
Frequency of appearance of individual bands (wavelengths) in the eight discriminant models.

**Figure 2 insects-12-00047-f002:**
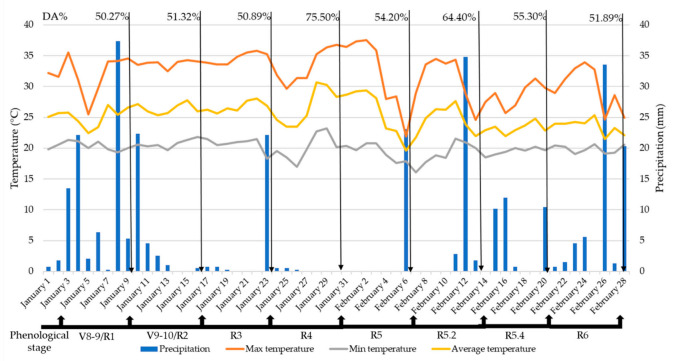
Meteorological data and discriminant analysis (DA) accuracy (%). Letter “V” stands for vegetative stages and “R” for reproductive stages.

**Figure 3 insects-12-00047-f003:**
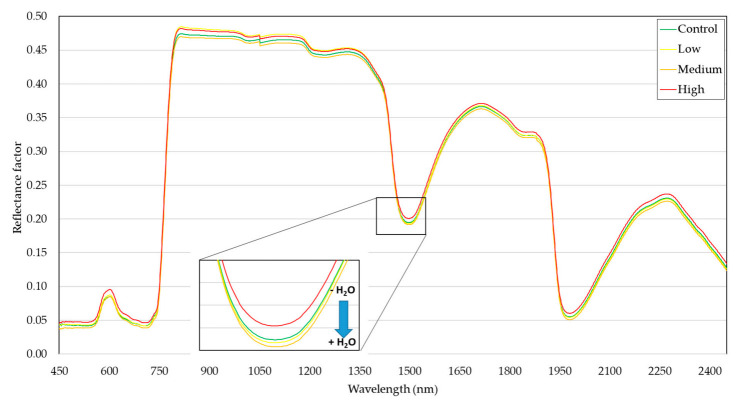
Average spectral curves (450–2400 nm) of soybean leaves under different levels of whitefly infestation.

**Figure 4 insects-12-00047-f004:**
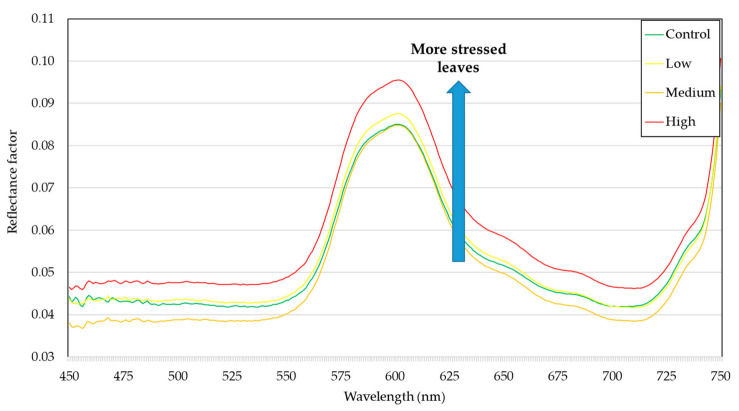
Average spectral curves (450–750 nm) of soybean leaves under different levels of whitefly infestation.

**Figure 5 insects-12-00047-f005:**
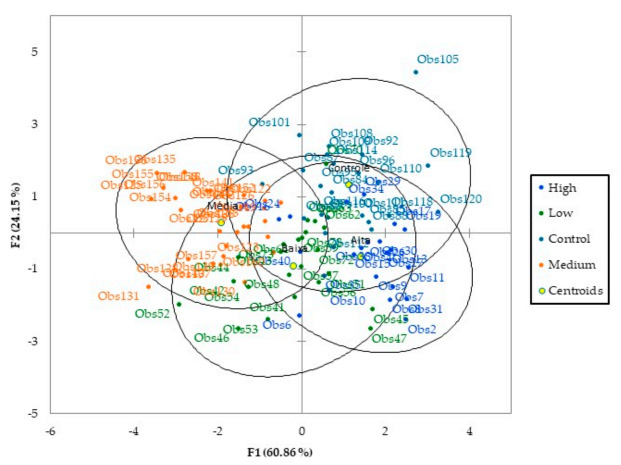
Discriminant analysis of hyperspectral data (450–2400 nm) of soybean leaves under different levels of whitefly infestation. Obs stands for observation, and F1 and F2 for function.

**Table 1 insects-12-00047-t001:** Linear discriminant classification of hyperspectral data (450–2400 nm) of whitefly infested soybean leaves (n = 160). Independent validation was carried out with 48 samples and classified with 75.48% accuracy.

Actual Class	Assigned Class by Training Model			
	High	Low	Control	Medium
High	17 (73.91%)	3	2	1
Low	2	16 (69.57%)	3	2
Control	1	3	16 (72.73%)	2
Medium	0	4	0	24 (85.71%)

## Data Availability

The data presented in this study are available on request from the corresponding author.

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
