# Peer review of "Monitoring Bemisia tabaci (Gennadius) (Hemiptera: Aleyrodidae) Infestation in Soybean by Proximal Sensing"

_insects, 2021, doi:10.3390/insects12010047_

Round 1

Reviewer 1 Report

The manuscript presents a study on the use of spectral profiles to detect stress caused by whiteflies in soybean crops. The analysis of the spectral curves described in the article is interesting and potentially useful, but there are a few weaknesses that need to be properly addressed, as detailed below.
- The language used in the manuscript is generally intelligible, but there are numerous minor problems that could be eliminated by a careful revision by someone more familiarized with the English language.
- The two first paragraphs of the introduction have a lot of redundancy and pieces of information that are not relevant in the context of this work. The paragraphs could easily be condensed into two or three sentences.
- Lines 86-91: the collection of spectral data is not a geotechnology. Mentioning the term “geotechnology” only introduces confusion to the text and should be avoided.
- Line 99: I am not sure this statement is correct. There are plenty of studies that use wavelengths in the 1000nm – 2000nm. The wavelengths around 1400nm are particularly important in many cases, as it is one of the water’s absorption bands.
- The contextualization in the introduction section is lacking. There have been many studies applying hyperspectral sensing (including hyperspectral images) for the detection of stresses. The authors should briefly present the major approaches used in the literature, and then emphasize the main contributions of their investigation by either revealing how their method improves upon its predecessors, or by highlighting the new findings produced by the experiments. As is, the contribution of the manuscript is not clear at all.
- Line 131, 138: Date formats vary considerably around the world, please use a less ambiguous format (e.g. December 13, 2018).
- Lines 171, 174: “Information” is an uncountable term, thus you cannot use “large number of information”. You should use “large amount of information”, or something equivalent.
- The results and discussion section includes many useful remarks, but one very important aspect is completely absent. Spectral curves can be very sensitive to several factors, including leaf’s age and the presence of water, nutrition and disease related stresses. In this context, there are two things that should be made clear in the text. First, the authors should indicate what kind of measures were taken in order to guarantee that only the stress caused by the whitefly was present (assuming that such measures were even considered). Second, the authors should emphasize that under real conditions, different types of stresses can be present. This is by far the greatest challenge of using spectral curves to detect stresses: these are very effective indicating that there is some kind of stress present, but discriminating among them has been proven to be extremely difficult to accomplish. Many authors maintain that the only way to achieve this goal is by fusing different types of data, which by itself is a challenging objective and currently a hot topic of research.

Author Response

The manuscript presents a study on the use of spectral profiles to detect stress caused by whiteflies in soybean crops. The analysis of the spectral curves described in the article is interesting and potentially useful, but there are a few weaknesses that need to be properly addressed, as detailed below.

- The language used in the manuscript is generally intelligible, but there are numerous minor problems that could be eliminated by a careful revision by someone more familiarized with the English language. – The manuscript was revised by an English native speaker.

- The two first paragraphs of the introduction have a lot of redundancy and pieces of information that are not relevant in the context of this work. The paragraphs could easily be condensed into two or three sentences. – First paragraph was adapted to a shorter version (Line 45) and the second was adapted and attached to the first (Line 50).

- Lines 86-91: the collection of spectral data is not a geotechnology. Mentioning the term “geotechnology” only introduces confusion to the text and should be avoided. - The authors appreciated the suggestion. The use of the term geotechnology had the intention of including different data collecting methods and their applications. In order to make it clearer we changed it to “proximal sensing”, which includes hyperspectral (Line 73).

- Line 99: I am not sure this statement is correct. There are plenty of studies that use wavelengths in the 1000nm – 2000nm. The wavelengths around 1400nm are particularly important in many cases, as it is one of the water’s absorption bands. - The reviewer is correct, but the authors had made research of the last articles published about this subject, and the majority of them using wavelengths in the region of visible and red-edge, that are directly influenced by pigments (chlorophyll, carotenoids, and xanthophyll) and photosynthetic process. To make it clearer we changed it to 400 – 2500 nm, which is the region currently used more often by remote sensing regarding vegetation (Line 86).

- The contextualization in the introduction section is lacking. There have been many studies applying hyperspectral sensing (including hyperspectral images) for the detection of stresses. The authors should briefly present the major approaches used in the literature, and then emphasize the main contributions of their investigation by either revealing how their method improves upon its predecessors, or by highlighting the new findings produced by the experiments. As is, the contribution of the manuscript is not clear at all. – We included more information regarding hyperspectral sensing in the Introduction section for better contextualization and to show how our study contributes to this subject. The references 36 and 37 were added (36 - Iost Filho et al. 2020 and 37 – Alves et al. 2019).

- Line 131, 138: Date formats vary considerably around the world, please use a less ambiguous format (e.g. December 13, 2018). – The date formats were adapted to e.g. Nov 28, 2018.

- Lines 171, 174: “Information” is an uncountable term, thus you cannot use “large number of information”. You should use “large amount of information”, or something equivalent. – “Information” was adapted to “data”.

- The results and discussion section includes many useful remarks, but one very important aspect is completely absent. Spectral curves can be very sensitive to several factors, including leaf’s age and the presence of water, nutrition and disease related stresses. In this context, there are two things that should be made clear in the text. First, the authors should indicate what kind of measures were taken in order to guarantee that only the stress caused by the whitefly was present (assuming that such measures were even considered). Second, the authors should emphasize that under real conditions, different types of stresses can be present. This is by far the greatest challenge of using spectral curves to detect stresses: these are very effective indicating that there is some kind of stress present, but discriminating among them has been proven to be extremely difficult to accomplish. Many authors maintain that the only way to achieve this goal is by fusing different types of data, which by itself is a challenging objective and currently a hot topic of research. – To minimize the impact of these factors we standardized the section of the plant that the leaves were collected from (middle third of the soybean plants) additionally we used the randomized block design and four repetitions. The plants in each repetition were also isolated inside the cages to avoid the interference of other pests. These in field conditions were implemented to control the environmental effects.

Reviewer 2 Report

This type of study is essential to develop the IPM system and has merits for publication. Nonetheless, this manuscript is not acceptable in the present form because several shortcomings are found.

Most serious problem is redundancy. For example, introduction section can be shorter. In L43-56, L62-79, this kind of information is remotely related the subject of this manuscript. One more example, in L233-234, I don’t know the relationship of "the information of first outbreak" to the subject of this MS. Please check all text and revise this shortcoming.

In addition, I think some figures will be hard to read caused by font size if this manuscript will be published as the PDF. Please revise this matter. I will point out the detail below.

L123-124: Commonly, fertilization affect the insect damage. Therefore, please describe the detail if it possible.

Figure 2: Please reconsider font size, especially on “Phenological stage”.

Figure 3,4: Please reconsider font size of “Vertical and Horizontal axis” and characters in the figure.

Author Response

This type of study is essential to develop the IPM system and has merits for publication. Nonetheless, this manuscript is not acceptable in the present form because several shortcomings are found.

Most serious problem is redundancy. For example, introduction section can be shorter. - We appreciate the suggestion, that brings more clarity and objectivity to the introduction, and we removed disconnected paragraphs (first paragraph was adapted to a shorter version – Line 45).

L43-56, L62-79, this kind of information is remotely related the subject of this manuscript. One more example – The paragraphs have been adapted from lines 45 to 83. Parts of the sentences were removed making it more concise.

L233-234, I don’t know the relationship of "the information of first outbreak" to the subject of this MS. Please check all text and revise this shortcoming. – The sentence was adapted to “High densities of B. tabaci causes the occurrence of honeydew, in the nymphal stage, it excretes a high volume of this sugar-rich watery fluid” for better understanding.

In addition, I think some figures will be hard to read caused by font size if this manuscript will be published as the PDF. Please revise this matter. I will point out the detail below. - This suggestion was presented and accepted in the first review. The authors used the standard font size of the journal, but we increased all the figures font sizes to make it easier for reading and interpretation of our results.

L123-124: Commonly, fertilization affect the insect damage. Therefore, please describe the detail if it possible. - The authors understand the importance of this suggestion and included clearer information regarding fertilization. (Line 122-123).

Figure 2: Please reconsider font size, especially on “Phenological stage”. - Figure adapted from font size 12 to 14.

Figure 3,4: Please reconsider font size of “Vertical and Horizontal axis” and characters in the figure. - Figures were adapted to font size 14 in axes and characters.

Round 2

Reviewer 1 Report

My main coincerns were properly addressed.

Reviewer 2 Report

This MS was revised well.

This manuscript is a resubmission of an earlier submission. The following is a list of the peer review reports and author responses from that submission.

Round 1

Reviewer 1 Report

I made some comments and asked some question in the PDF file for authors to consider addressing. MS can be improved for reader to follow and understand what and how (and why) things were done in a clear sequence, which i believe this indeed would be very useful improvement. I hope my suggestions are helpful for authors to revise their MS.

Reviewer 2 Report

This manuscript described the spectral responses of soybean plants to whiteflies. The authors created a gradient of injury by artificially infesting the plants with different numbers of adults. Although it is early to conclude about field applications, the results seem promising for detecting Bemisia tabaci on soybean using remote sensing. This is not a novel research paper. Similar and more robust results can be found at Yang, Chenghai, and James H. Everitt. "Remote Sensing for Detecting and Mapping Whitefly (Bemisia tabaci) Infestations." The Whitefly, Bemisia tabaci (Homoptera: Aleyrodidae) Interaction with Geminivirus-Infected Host Plants. Springer, Dordrecht, 2011. 357-381.

In addition, the authors seemed to have published a pretty similar manuscript somewhere else: Barros, Pedro PS, et al. "Monitoring Bemisia tabaci Gennadius (Hemiptera: Aleyrodidae) infestation in soybean using hyperspectral remote sensing." Preprints (2020).

Anyway, I provided some other questions to consider:

  1. 26: What is the difference between abstract and simple abstract? Please refer to journal’s guidelines.
  2. 42-92: It is strange the construction of the paragraphs. Seems to be that need throughout revision of what makes a good sentence vs. a good paragraph.
  3. 51: What is the criteria used to determine most important crop? Dollars? Did you consider the entire agricultural chain?
  4. 58: EIL is governed by cost of management (as stated), but also market value, injury units per pest, damage per injury unit, and the proportional reduction in pest attack. Please refer to Stern et al. and Pedigo et al. for a detailed definition.
  5. 91: This study aimed…

L 98-100: Is the climate information important for interpreting your study? Why not including these information mentioned here in the results then?

  1. 109 and 110: Please use vocabulary consistently: bioassay vs. assay
  2. 110: Is V3 the same as in three fully expanded trifoliates?
  3. 112 and L. 117: What was the criteria used to determine low, medium, and high infestation? How long the adults stayed feeding? Did you consider cumulative insect injury through time?
  4. 160: Are you referring to the infestation in your study or in general? What do you mean by alarming?
  5. 160: Only high densities (>1200 adults) cause losses of 30%. Do low and medium densities also cause yield losses?
  6. 160-175: Much of this part can be moved to the introduction. Please focus in the discussion the results obtained from your study.

Figure 1 and 2 need bigger font sizes. What is DA%?